# Fucosidosis—Clinical Manifestation, Long-Term Outcomes, and Genetic Profile—Review and Case Series

**DOI:** 10.3390/genes11111383

**Published:** 2020-11-22

**Authors:** Karolina M. Stepien, Elżbieta Ciara, Aleksandra Jezela-Stanek

**Affiliations:** 1Adult Inherited Metabolic Diseases, Salford Royal NHS Foundation Trust, Salford M6 8HD, UK; 2Department of Medical Genetics, The Children’s Memorial Heath Institute, 04-730 Warsaw, Poland; e.ciara@ipczd.pl; 3Department of Genetics and Clinical Immunology, National Institute of Tuberculosis and Lung Diseases, 01-138 Warsaw, Poland; jezela@gmail.com

**Keywords:** fucosidosis, *FUCA1*, lysosomal storage disease, neurocognitive dysfunction, clinical outcome, genotype

## Abstract

Fucosidosis is a neurodegenerative disorder which progresses inexorably. Clinical features include coarse facial features, growth retardation, recurrent upper respiratory infections, dysostosis multiplex, and angiokeratoma corporis diffusum. Fucosidosis is caused by mutations in the *FUCA1* gene resulting in α-L-fucosidase deficiency. Only 36 pathogenic variants in the *FUCA1* gene are related to fucosidosis. Most of them are missense/nonsense substitutions; six missense and 11 nonsense mutations. Among deletions there were eight small and five gross changes. So far, only three splice site variants have been described—one small deletion, one complete deletion and one stop-loss mutation. The disease has a significant clinical variability, the cause of which is not well understood. The genotype–phenotype correlation has not been well defined. This review describes the genetic profile and clinical manifestations of fucosidosis in pediatric and adult cases.

## 1. Introduction

Fucosidosis (OMIM# 230000) is a lysosomal storage disease (LSD) caused by biallelic pathogenic variants in *FUCA1* gene localized on chromosome 1p36.11 [1,2,3]. Mutations result in α-L-fucosidase (EC 3.2.1.51) deficiency [4] in almost all patients [5]. The enzyme is a homotetramer composed of subunits of different masses (50 to 60 kDa), which result from variations in N-glycosylation and proteolytic processing [6,7]. As a result of the hydrolytic enzyme deficiency, incomplete catabolism of N- and O-glycosylproteins results in the accumulation of fucose-containing glycolipids and glycoproteins in various tissues and urine [7]. In particular, glycolipids expressing blood group antigens (H, X) accumulate in the liver. 

Fucose (C_6_H_12_O_6_), a deoxy sugar isolated both as free sugar and as a component of oligosaccharides in human and sheep milk, is found in most of the plasma glycoproteins in the mucopolysaccharides and mucolipids of various human and animal tissues, where it is present mainly as a terminal sugar [8]. The impaired hydrolysis of terminal α-L-fucoside linkages in glycosphingolipids and glycoproteins leads to accretion of fucose-containing oligosaccharides, glycoproteins, and glycolipids in the brain, liver and skin [1,2]. Oligosaccharides and glycoasparagines are deposited in tissues and excessively excreted in urine.

Patients affected with fucosidosis excrete a large amount of glycopeptide in the urine [7], which is typified by the presence of an α1-6-linked fucose linked to the chitobiose core. A glycoasparagine Fuc (α1-6)GlcNAcBeta1-Asn, the main glycopeptide, presented in urine, matches to the linkage region of the oligosaccharide [9]. The large amount of glycopeptide excreted in fucosidosis can result from the steric inhibition of the glycosylasparginase by the fucose residue. The non-fucosylated glycoasparagines are normally processed with the accumulation of only fucosylated glycoasparagine and oligosaccharides, which lack α1-6-linked fucose residue [10]. 

Among glycoproteins that accumulate in fucosidosis are saposins. They are sphingolipid activator proteins derived from a single precursor, prosaposin, by proteolytic processing [11]. In particular, saposins A and D accumulate in liver in patients with fucosidosis and play no apparent role in the reduction of storage material. Although the mechanism is uncertain, the massive storage of saposins A and D was thought to be key in the pathogenesis of fucosidosis [11]. 

The first case of fucosidosis was described by Durand et al. [12]. Fucosidosis was initially known as mucopolysaccharidosis type F [8,12,13,14].

### Incidence

Fucosidosis is an orphan disease with a very low incidence of <1/200,000 [15]. So far approximately 120 cases have been described [16,17,18]. The highest incidence has been described in Italy, in the Hispanic-American population of New Mexico and Colorado, and in Cuba [16,19]. 

## 2. Diagnosis

### 2.1. Enzymatic Activity

The diagnosis of fucosidosis is made by measuring the enzyme activity. The peak L-fucosidase activity in serum and plasma is attained at pH 5.0 [20], and in dry blood spot card (DBS) samples was found to be linear at different time intervals over a period of 20 h [21]. 

### 2.2. Genetic Profile

Fucosidosis is an autosomal recessive disorder induced by biallelic mutations in the structural gene of α-L-fucosidosis (*FUCA1*). A patient may have two copies of the same mutation (homozygote) or two different mutations (compound heterozygote). 

The α-L-fucosidase *FUCA1* gene (*612280) is localized to chromosome 1p36.11 and contains eight exons covering 23 kb [22]. It encodes an unprocessed 461 amino acids protein—22 amino acids of the signal peptide and 439 of the mature protein [23,24].

Only 36 pathogenic variants in the *FUCA1* gene reported in HGMD are so far associated with fucosidosis (www.hgmd.cf.ac.uk; updated 10 November 2020) [25] (Table 1).

Most of them are missense/nonsense substitutions, in which there are six missense and 11 nonsense mutations. Deletions occur frequently, including eight small and five gross changes. So far, only three splice site variants have been described—one small deletion, one complete deletion, and one stop-loss mutation [25].

All pathogenic variants result in barely detectable α-L-fucosidase activity and significantly deficient cross-reacting immune material (CRIM; below 6% of normal mean), implying that the mutant fucosidase proteins are unstable and degrade at a great rate [1,5,30]. 

A pseudogene *FUCA1P* was found on chromosome 2q31–q32. It shares 80% identity with α-L-fucosidase gene and contains no intron as well as open reading frame [22]. A locus on chromosome 6q24.2 (*FUCA2*) has been shown to influence plasma fucosidase activity (representing 10–20% of the total cellular fucosidase activity), but not its activity in leukocytes. This gene is not involved in fucosidosis [49].

### 2.3. Molecular Testing

Molecular genetic diagnostics include a mix of *FUCA1*-gene-targeted testing and chromosomal microarray analysis depending on the molecular background of the disease. *FUCA1* sequence analysis detects most of the known pathogenic changes—intragenic small deletions/insertions (INDELs) and missense, nonsense, and splice sites variants. Gross deletions/duplications (exons or the whole gene) are detected by chromosomal microarray analysis (CMA). The sequence analysis should be performed first. If only one or no pathogenic variant is found, array comparative genomic hybridization (aCGH) analysis is recommended.

*FUCA1*-gene-targeted analysis requires that the clinician determine that fucosidosis is the most probable disease in patient, whereas multigene panel testing proves to be valuable in making differential diagnoses. Because the clinical manifestation of fucosidosis is broad ranging, individuals with the distinctive phenotype may successfully be diagnosed using single-gene testing, while those in whom the clinical diagnosis is unclear need verification using more comprehensive testing, including whole-exome sequencing.

### 2.4. Genotype–Phenotype Correlation

Since most affected individuals are homozygotes for different mutations, strict genotype–phenotype correlation is not well defined. Only few of them are repeated in unrelated patients. As a rule, patients are homozygous for two null alleles, such as the gross deletions, including G33 or G34 genotype (Table 1 and Table 2), have a severe phenotype [1,16].

The clinical heterogeneity is less commonly associated with different non-allelic mutations, as complementation studies between type I and type II fucosidosis (see below) did not restore α-L-fucosidase activity [51]. The possible existence of different co-allelic mutations in the fucosidase gene, responsible for different clinical phenotypes, has not yet been investigated [5]. This hypothesis however disagrees with the existence of different clinical phenotypes within the same family [52]. The clinical variability of fucosidosis is broad and it was proposed that other genetic or non-genetic factors may take part in it [7].

### 2.5. Prenatal Diagnosis 

Fucosidosis is inherited in an autosomal recessive manner, which means that each sibling of the patient from the same biological parents has a 25% risk of being affected, a 50% chance of being an heterozygous healthy carrier, and a 25% chance of being healthy and not a carrier [1]. 

Once the *FUCA1* pathogenic variants have been identified in an affected family member, parents may decide on prenatal testing in the subsequent pregnancy or preimplantation genetic testing (PGT-M). 

Only molecular genetic testing can be used. Biochemical testing is not useful because the expression level is nearly 50% lower in mutation carriers than in the normal control [36]. The Western blot analysis showed that mutation caused the decrease of protein *FUCA1* expression level and mutation carriers still had partial protein expression, but lower than normal controls [36].

## 3. Clinical Manifestation 

Fucosidosis is a neurodegenerative disorder which progresses inexorably. Its features, such as coarse facial features, growth retardation, recurrent upper respiratory infections, dysostosis multiplex, and angiokeratoma corporis diffusum are common in other LSDs. The mean age at presentation of the first symptom in 60 patients was 1.2 +/− 0.8 years (range: 0.0–5.5 years) [1].

### 3.1. Type I and II

Although the first case of fucosidosis was described in 1966 [12], the enzyme deficiency was only characterized in 1968 [53].

In view of the significant variability in clinical manifestations in fucosidosis, two main types have been defined [54,55]. Type I: a rapidly-progressing neurodegenerative course, leading to decerebration and death typically before the age of 10 years [13];Type II: a milder course, with slower neurological symptoms progression, possible survival into adulthood, and most patients develop angiokeratoma corporis diffusum [14].

### 3.2. Skin

Dermatological abnormalities were documented in 60% of patient and included telangiectasiae on the skin or conjunctivae and angiokeratoma [1]. Angiokeratoma corporis diffusum was present in most (51%) patients [1], particularly in those living ten years and longer, although it is not pathognomonic for fucosidosis, it raises a suspicion of the diagnosis [56]. The older the patient is when examined, the more likely that angiokeratomas are found [57]. 

Red, purple pinhead-sized raised skin lesions usually develop on the lower abdomen and genitalia, while tiny scattered angiomas cover most of the body [1]. Angiokeratomas, within the papillary dermis, are proliferative ectatic blood vessels limited by a flattened endothelium containing erythrocytes [58]. Some patients have only telangiectasia without angiokeratoma [18]. 

Apart from angiokeratoma corporis diffusum, skin abnormalities also include widespread telangiectasia, skin thickness, hyperhidrosis and hypohidrosis, acrocyanosis, and distal transverse nail bands [1,59].

Valero-Rubio et al. [60] showed that *FUCA1* expression disturbances, resulting from its down-regulation, affect mainly genes related to keratinocyte differentiation/epidermal development and immune responses. This might have been instigated by abnormal transcription factor expression (FOXN1). The authors thus postulated that skin lesions in fucosidosis might be caused by dysfunctions in common aetiological overlapping molecular cascades [60].

### 3.3. Eye

Dilated and tortuous retinal veins were observed in 54%, dilated and tortuous conjunctival vessels in 53%, microaneurysms of conjunctival vessels in 41%, corneal opacities in 11%, and pigmentary retinopathy in 7% of patients and vision loss in 6% of patients [1]. Severe visual impairment is uncommon. 

It was shown that storage material accumulates in conjunctival, retinal, and skin vessels. Histological assessment of the conjunctiva’s endothelial cells has shown two distinct types of vacuoles—clear ones with a reticular structure similar to those seen in the mucopolysaccharidoses and dark inclusions with a dense granular material [61,62].

Blepharospasm is common. Both bulbar and, to a lesser extent, palpebral conjunctivae, have dilated tortuous vessels, some with saccular dilatations [63].

The anterior chambers are of normal depth with clear cornea and lenses. The fundi show macular changes similar to bull’s eye retinopathy; the foveal area has fine brown pigmentation ringed by a zone of hypopigmentation. The rest of the macula has fine brownish and/or, slatey-blue pigmentary lesion with occasional clumps of dark brown discoloration. No undue tortuosity of the vessels are present, the discs and vessels are normal.

Snodgrass et al. [63] and Gatti, et al. [54] reported ‘slight cloudiness of corneae’ in some cases [64]. Borrone et al. [64] reported two affected siblings of whom one presented with corneal clouding, strabismus, papilledema with congestion, and tortuosity of retinal vessels, and the other with thin and tortuous vessels.

Another 14-year-old female patient presented with a slight limitation in right abduction, an 18-prism diopter exophoria in primary gaze at distance with a latent component fine jerky vertical nystagmus. She had 100 s of arc stereopsis. On slit lamp examination telangiectatic vessels in the inferior conjunctiva and mild corneal stromal haze bilaterally proportional to her slightly decreased best corrected visual acuity were noted [65]. 

### 3.4. Facial Features/Anthropometrics

Coarse facial features are common [65,66,67] and may resemble gargoylism [63]. There are however cases with no obvious facial coarsening [68]. 

The height of 91% of patients was reported to be below the 5th centile [1], while the weight of 77% was below the fifth centile.

Occipitofrontal circumference (OFC) was shown to be normal (2nd–98th centile) in 79% of patients. 15% were microcephalic (OFC < 2nd centile) and 4% of patients were macrocephalic (OCF > 98th centile) [1]. The skull was large in a case reported by Snodgrass [63].

### 3.5. Cardiac Manifestations

Cardiomyopathy is not a common feature [18] but mild mitral regurgitation was found in 50% of patients investigated [69]. An enlarged heart with left ventricular hypertrophy and dilatation of right cavity can be observed on an X-ray and the electrocardiogram may show incomplete right bundle branch block [8]. Cloudy degeneration of myocardium and stasis in coronary vessels was also previously observed [8]. 

### 3.6. Respiratory/Recurrent Infections

Up to 78% of patients experience respiratory tract infections [1]. Intermittent otitis media, upper respiratory tract infections, and breathing difficulties during sleep as a result of partial upper airway obstruction were documented [70]. Moreover, a recurrent respiratory infection every 1 or 2 months, despite normal immunological tests [18] and diffuse emphysema with little areas of atelectasis of the lungs was previously observed [8]. 

Peripheral blood smears confirmed the presence of cytoplasmic vacuoles in many lymphocytes which were only slightly periodic acid-Schiff (PAS) positive [8].

Possible links between fucosidosis and cystic fibrosis were postulated [71]. High sweat electrolytes and recurrent infections of the respiratory tract are features of both disorders. Fucosidosis patients often have recurrent infections confined to areas of mucus-secreting ciliated epithelia but their immune system was shown to be intact and therefore it was suspected that there could be a local defect in mucus clearance. [71]. 

It was proposed that the terminal sugars fucose and sialic acid play a major role in defining the viscoelasticity of mucus. Thus, alterations in the enzymatic cleavage of these sugars affect mucus cross-linking and its viscoelasticity. Without cross-linking, cilia would flail about ineffectively in watery secretions [71].

### 3.7. Hepatosplenomegaly 

Hepatomegaly was documented in 20% [69] to 40% of patients [1] and was not progressive. Splenomegaly was documented in 25% of patients [1]. The liver was often palpable 3 cm to 5 cm (68, 72] and the spleen 1 cm to 3 cm below costal margins [68,72] 

In some cases, hepatomegaly was not apparent [18,57,73]. In rare cases of fucosidosis type II, massive splenomegaly may be related to portal cavernoma [69]. 

Liver biopsy may reveal hepatic cells containing abundant stored ceramide tetra- and pentahexoside, producing a clear, swollen appearance of balloon or pseudogargoyle cells which are only weakly PAS positive [8,53]. 

### 3.8. Dysostosis Multiplex

Some patients also have non-specific features of dysostosis multiplex [74], which may affect up to 58% of patients [1]. An X-ray of the spine shows small thoracolumbar vertebrae with anterior tonguing, short odontoid pegs, cervical platyspondyly, and broad ribs [75]. Moreover, prognathism, ribs deformities, scoliosis with partial fusion of lower thoracic vertebrae, and gibbous deformity of lumbar vertebrae have been described, as well as absent scaphoid, trapezium, and trapezoid bones in the wrists [63]. Joint contractures have also been a common feature [30]. 

### 3.9. Neurology/Neurocognitive Function

Willems et al. (1991) documented that up to 95% of patients have progressive mental health deterioration and up to 87% have motor deterioration, while seizures may occur in 38% of patients [1]. Forty-one percent of the patients lost the ability to sit, 38% lost the ability to stand, 50% became unable to walk alone, and 67% lost the ability to speak. Twenty-eight percent of the patients had a rapid neurologic deterioration with complete loss of the ability to walk, stand, sit, or talk before the age 5 years. Slower neurologic deterioration and maintenance of the ability to sit, stand, walk, or talk after the age of 10 was documented in 53% of the patients. In the remaining 19%, the course of neurologic deterioration was intermediate. 

Psychomotor decline was noted in 60% of patients at a median age of 18 ± 4.5 months [69].

Progressive neurological degeneration results in flexion contractures of legs and arms [63]. Spastic quadraparesis with increased deep tendon reflexes was previously described [19,68,73] and estimated to affect up to 40% of all fucosidosis cases [47]. Gradually-increasing spasticity with bilaterally-increased deep tendon reflexes and unsustained ankle clonus may result in unsteady gait, associated with excessive femoral anteversion, tibial torsion, mild metatarsus adductus, and intoeing [70].

Seizure types may vary with the secondary generalized epilepsy reported before [70]. Seizure was a feature in several cases [67] and documented in up to 38% of all fucosidosis cases [47].

Focal dystonia is a rare neurological symptom [66,76,77] and described in 12% of fucosidosis cases [47]. 

Hearing is usually intact [63]. There are however cases with impaired hearing [65], asymmetric mild-moderate sensorineural hearing loss and Eustachian tube dysfunction [19]. 

### 3.10. Neuroimaging

Neuroimaging features are commonly described. There is evidence of hypomyelination, with extensive, confluent, progressive, and symmetric signal anomaly in the periventricular and subcortical white matter [28,66,70,78,79]. Apart from the white matter changes, the grey matter changes were described in fucosidosis and are characterized by signal changes in the globi pallidi [15,28,66,78,79,80]. This unique-for-fucosidosis feature manifests as marked hypointensity on T2/FLAIR sequences, and hyperintensity on T1 sequences [28,66,76,79]. A hypointense area on T2-weighted imaging in the bilateral globus pallidus may indicate the presence of high levels of iron in echo-gradient MRI [73], and, in combination with curvilinear T2-hyperintense areas within lentiform nuclei, creates a sign sometimes called “eye of the tiger” [18].

Generalized cerebral and cerebellar atrophy was also observed on CT scan with the disease progress [19,28,66,67,78] and was more commonly documented in patients with the type II phenotype [78]. Among patients who survived beyond the age of 30, cranial CT abnormalities included infra- and supratentorial volume loss, in particular in the frontal lobes [81].

Ventricular dilatation and focal areas of hypodensity were also observed [82]. Kau et al. [83] noted an increased cerebellar volume in the early stage of fucosidosis. The prominent white matter abnormalities and low signal of the globus pallidus may help distinguish it from other neurometabolic disorders [39,78].

Another modality, MS spectroscopy, typically shows a decreased N-acetyl aspartate (NAA)/choline ratio and a characteristic abnormal peak at 3.8 ppm [28,66] and a double peak at 1.2 ppm [80]. The level of N–acetylaspartate and creatinine ratio was shown be decreased with an unusual lactic acid peak at 1.33 ppm in MRI spectra in the bilateral basal ganglia and posterior limb of internal capsule [67]. In patients who had undergone hematopoietic stem cell transplantation (HSCT), magnetic resonance spectroscopy (MRS) confirmed the disappearance of an unusual lactic acid peak at 1.33 ppm, and the rise of the of NAA and creatinine ratio to normal [67].

### 3.11. Mortality

Death before age 10 years occurs in less than half patients (43%) and after age 20 similarly it occurs in 41% [1]. Patients with early symptoms tend to show faster neurologic deterioration leading to mortality at young age [1]. It was documented that around 60% of patients die secondary to respiratory infections and neurological deteriorations [69]. 

### 3.12. Animal Model

A Fuca1-deficient mouse model was generated by gene-targeting techniques [84]. The authors demonstrated behavioral alterations in Fuca1-deficient mice that coincided with early signs of neuropathology (at 3 months of age); subtle abnormalities in sensorimotor and cognitive abilities were identified. The animals displayed lysosomal dysregulation (increased Lamp1 expression) and evidence of neuroinflammation and secondary storage of GM2 ganglioside, as early manifestations of brain pathology [85]. These abnormalities preceded neuron loss but imply an early stage of neuropathology encompassing an extended endosomal-lysosomal network, secondary lipid storage, and emerging microgliosis and astrogliosis [86]. 

## 4. Therapeutic Options

### 4.1. Supportive and MDT

The cornerstone of management is generally supportive with physiotherapy and other allied health input. The multidisciplinary team usually involves pediatric and adult metabolic physicians, metabolic physiotherapist, and ophthalmology, orthopedic, and cardiology specialists. In case of any neurological/neuropsychiatric complications, input from a neurology, neuropsychiatry teams should be sought, with a full neuropsychology assessment. 

### 4.2. Haematopoietic-Stem-Cell Transplantation

Haematopoietic-stem-cell transplantation (HSCT) showed promising results in canines with fucosidase deficiency; α-L-fucosidase enzyme activity normalized in leucocytes, plasma, and neural and visceral tissues [86,87]. 

HSCT was applied in a small number of patients with fucosidosis with symptoms stabilization in some cases. It was first attempted in an asymptomatic 7-month-old child who was diagnosed with fucosidosis after the diagnosis of the symptomatic older brother [88]. After it was reported as an effective treatment, it was performed very uncommonly, and only after careful patient selection [3,89]. Earlier transplant appeared to be more effective in fucosidosis than transplant after symptoms fully manifest. After successful transplantation, a gradual increase in α–L fucosidase levels was observed in plasma, white blood cells, and cerebrospinal fluid, as well as an improvement in myelination on MRI scan for up to 4 years after transplantation [88,89].

In a case described by Milano et al. [89] at four years post-HSCT, there were major improvements in MRI findings, psychomotor development, swallowing, and number of respiratory tract infections. In a case of a symptomatic 3-year-old girl who underwent an umbilical cord blood (UCB) HSCT, a remarkable regression of neurological symptoms after HSCT was detected, cranial MRI results at two years post-HCT improved, a prior lactic acid peak disappeared on MRS, but no impact on dysostosis multiplex was observed [67]. Two out of three other patients showed decelerating of neurologic progress and a dramatic reduction in respiratory infections after HSCT [90,91]. Other HSCT-related trials were completed in 2008 [92].

### 4.3. Enzyme Replacement Therapy 

Enzyme replacement therapy (ERT) for fucosidosis is currently being tested in preclinical studies. It is the first glycoprotein disorder to use intracisternal ERT to deliver a recombinant enzyme directly to the central nervous system (CNS) in a canine [93]. It was confirmed that the enzyme was higher near the injection site (39–73% of normal) and lower in deep brain structures (2.6–5.5% of normal), but it reached all areas of the CNS. The direct CNS delivery of the enzyme was shown to be safe and in a follow-up study (in the same experiment) partial improvement in neuropathology following intracisternal ERT has been shown, implying that perhaps a more intensive ERT protocol is required to improve outcomes [91,93].

Gene therapy may be an alternative treatment [94,95] but no clinical trials are available yet. 

## 5. Discussion 

Atypical presentation of fucosidosis may lead to delay in diagnosis, in particular in absence of coarse features or hepatosplenomegaly or joint contractures. As an example, a female patient aged 4 years developed swollen, painful knees, hips, wrists, hands and feet. While the inflammatory response was suggestive of polyartricular juvenile inflammatory rheumatoid arthritis [75], her symptoms were associated with the underlying metabolic disease. Diagnostic delay could be avoided with remembering the wide spectrum of clinical and radiological findings of fucosidosis with atypical presentations of disease.

Telangiectasia or angiokeratoma, despite it being inconspicuous in the early stage of the disease, might be its early indicator. Careful review of the classic symptoms, including visceromegaly, neurological deterioration, dysostosis multiplex, and neuroimaging, is important and allows for accurate diagnosis [18].

It is worth emphasizing that urinary screening of glycosaminoglycans (GAGs) does not detect this rare LSD, which requires specific enzyme analysis of plasma and leucocytes samples, followed by genetic analysis of *FUCA1* mutations. In addition, a negative urinary oligosaccharide screen, seen in attenuated phenotype, does not exclude a diagnosis of LSD [75].

Neuroimaging is nearly always abnormal when performed in patients with fucosidosis. In terms of diagnostic investigations, biochemical testing is the first-line test and the preferred method for confirming the diagnosis. However, with easier access to next generation sequencing (NGS), more cases of fucosidosis are expected to be recognized, and—in such circumstances—the traditional pathway of biochemical testing will serve for verification. Given a potential benefit from transplantation in the early stages of disease, this is a potentially-treatable disorder [3,89]. Unfortunately, individuals are often diagnosed with advanced organ involvement and thus considered unsuitable for treatment with transplantation.

Genetic testing may be the only way to diagnose fucosidosis before the full spectrum of clinical manifestations will disclose. The genotype–phenotype correlation, as for other autosomal recessive disorders is, however, unclear and not straightforward. Given the genotypes summarized in Table 1, deletions, causing loss of enzyme activity (G7.2, G33, and G34) and variants leading to protein truncation, usually result in decreasing enzymatic activity (G2–G7.1, G9.1, G10.2, G11–G15, G17–G19, G22, G25–G29.2, G31, and G32.2) are expected to cause more severe fucosidosis manifestation. Unfortunately, the phenotypic descriptions are often lacking (Table 2). For variants altering the acceptor splice sites and available clinical records, we can conclude that G10.1 and G30 may result in later-onset disease and normal development in early childhood, with rapid progression from the age of three and cerebral atrophy in G10.1 [35] and regression from the age of two with very low, residual activity of α-L-fucosidase in G30 [47]. A severe course seems be the characteristic for G16, G20, G21, and G24, with observed psychomotor delayed from infancy in G24 [42], no activity of α-L-fucosidase in the leucocytes in G20 [40], and leukodystrophy described for G16 [37]. The conclusions need verification and further explanations of their pathomechanisms.

## Figures and Tables

**Table 1 genes-11-01383-t001:** *FUCA1* pathogenic variants causing fucosidosis phenotype reported to date.

Nucleotide Change ^2^	Amino Acid Change	Variant Type	Protein Domain	MAF	HGMD Accession	Zygosity Status	Genotype Set ^1^	Effect	References	PMID
c.203C > T	p.Ser68Leu	Missense	GHS	No data	CM940789	hmz	G1	The substitution exhibits a shift in polarity from polar to non-polar and displays an increase in Kyte–Doolittle hydrophobicity from −0.8 to 3.8. The variant occurs 399 amino acids from the end of the protein.	[26]	7874128
c.244C > T	p.Gln82*	Nonsense	GHS	<0.0001	CM930255	hmz; htz comp	G2 G3.1	This nonsense substitution truncates the protein at codon 82, which is 385 amino acids from the end of the protein.	[27,28,29]	8401503 26515723 8399358
c.355_364del10	p.(Glu119Thrfs*11)	Small del/fs	GHS	No data	CD972215	hmz	G4	This deletion results in a reading frame shift which truncates the protein at codon 130, which is 337 amino acids from the end of the protein.	[30]	9039984
c.393T > A	p.Tyr131*	Nonsense	GHS	<0.0001	CM022209	hmz	G5	This nonsense substitution truncates the protein at codon 131, which is 336 amino acids from the end of the protein.	[17,18,31]	12408193 17427030 32238081
c.437delC	p.(Pro146Argfs*41)	Small del/fs	GHS	No data	CD931119	hmz	G6	This deletion results in a reading frame shift which truncates the protein at codon 187, which is 280 amino acids from the end of the protein.	[27]	8401503
c.459G > A	p.Trp153*	Nonsense	GHS	No data	CM994353	htz comp	G7.1	This nonsense substitution truncates the protein at codon 153, which is 314 amino acids from the end of the protein.	[32]	10496076
c.464C > T	p.Ser155Phe	Missense	GHS	No data	CM144672	htz comp	G8.1	The substitution exhibits a shift in polarity from polar to non-polar and displays an increase in Kyte–Doolittle hydrophobicity from −0.8 to 2.8. The variant occurs 312 amino acids from the end of the protein.	[33]	24767253
c.467_468delAA	p.(Lys156Argfs*11)	Small del/fs	GHS	No data	CD931120	htz comp	G9.1	This deletion results in a reading frame shift which truncates the protein at codon 167, which is 300 amino acids from the end of the protein.	[34]	8504303
c.525-76_663-163del3282	p.?	Gross del	na	No data	CG1613053	htz comp	G10.1	This variant alters the acceptor splice site of exon 3 and the consequence of this change is not predictable, but a skip of exon 3 is very likely.	[35]	27706744
c.564G > A	p.Trp188*	Nonsense	GHS	No data	CM970537	hmz	G11	This nonsense substitution truncates the protein at codon 188, which is 279 amino acids from the end of the protein.	[30]	9039984
c.648C > A	p.Tyr216*	Nonsense	GHS	No data	CM930256	hmz	G12	This nonsense substitution truncates the protein at codon 216, which is 251 amino acids from the end of the protein.	[27]	8401503
c.661delA	p.(Ser221Alafs*6)	Small del/fs	GHS	<0.0001	CD930985	hmz htz comp ^3^;	G13 G14.1	This deletion results in a reading frame shift which truncates the protein at codon 227, which is 240 amino acids from the end of the protein.	[27,30]	8401503 9039984
c.671delC	p.(Pro224Leufs*3)	Small del/fs	GHS	No data	CD1613052	htz comp	G10.2	This deletion results in a reading frame shift which truncates the protein at codon 227, which is 240 amino acids from the end of the protein.	[35]	27706744
c.717C > A	p.Tyr239*	Nonsense	GHS	No data	CM1916910	hmz	G15	This nonsense substitution truncates the protein at codon 239, which is 228 amino acids from the end of the protein.	[36]	31603145
c.768+1G > A	p.?	Splice site	GHS	No data	CS1711597	hmz	G16	This substitution affects the invariant GT donor splice site of intron 4.	[37]	28097321
c.773delA	p.(Glu258Glyfs*3)	Small del/fs	GHS	No data	CD941679	hmz	G17	This deletion results in a reading frame shift which truncates the protein at codon 261, which is 206 amino acids from the end of the protein.	[38]	8081399
c.790C > T	p.Arg264*	Nonsense	GHS	<0.0001	CM144673	htz comp	G8.2	This nonsense substitution truncates the protein at codon 264, which is 203 amino acids from the end of the protein.	[33]	24767253
c.810delC	p.(Cys271Valfs*59)	Small del/fs	GHS	No data	CD93098	hmz	G18	This deletion results in a reading frame shift which truncates the protein at codon 330, which is 137 amino acids from the end of the protein.	[27]	8401503
c.837_838delTG	p.(Cys279*)	Small del	GHS	No data	CD184374	hmz	G19	This deletion results in a premature stop gain which truncates the protein at codon 279, which is 188 amino acids from the end of the protein.	[39]	29588375
c.969+1G > A	p.?	Splice site	GHS	No data	CS930812	hmz	G20	This substitution affects the invariant GT donor splice site of intron 5.	[40]	8097260
c.1000A > T	p.Asn334Tyr	Missense	GHS	<0.0001	CM970538	hmz	G21	The substitution does not exhibit a shift in polarity and displays an increase in Kyte–Doolittle hydrophobicity from −3.5 to −1.3. The variant occurs 133 amino acids from the end of the protein.	[30]	9039984
c.1003dupT	p.(Tyr335Leufs*9)	Small ins/fs	GHS	No data	CI972609	hmz	G22	This insertion results in a reading frame shift which truncates the protein at codon 344, which is 123 amino acids from the end of the protein.	[30]	9039984
c.1030_1095dup	p.(Asp344_Asn365dup)	Gross ins	GHS	No data	CN962970	hmz	G23	This in-frame variation leads to the duplication of 66bps (22 residues) in exon 6 and the consequence of this change is not predictable, but a loss of protein function is very likely.	[41]	8829645
c.1034G > A	p.Gly345Glu	Missense	GHS	No data	CM085971	htz	G24.1	The substitution exhibits a shift in polarity from non-polar to negatively charged and displays a decrease in Kyte–Doolittle hydrophobicity from −0.4 to −3.5. The variant occurs 122 amino acids from the end of the protein.	[42]	18504684
c.1060delA	p.(Arg354Glyfs*9)	Small del/fs	GHS	No data	CD19006	hmz	G25	This insertion results in a reading frame shift which truncates the protein at codon 363, which is 104 amino acids from the end of the protein.	[43]	30315573
c.1138G > T	p.Glu380*	Nonsense	FCT	<0.0001	CM920276	hmz	G26	This nonsense substitution truncates the protein at codon 380, which is 87 amino acids from the end of the protein.	[44]	1281988
c.1160G > A	p.Trp387*	Nonsense	GHS	<0.0001	CM930257	hmz; htz comp	G27 G9.2	This nonsense substitution truncates the protein at codon 387, which is 80 amino acids from the end of the protein.	[27]	8401503
c.1216G > T	p.Gly406*	Nonsense	FCT	<0.0001	CM950482	hmz	G28	This nonsense substitution truncates the protein at codon 406, which is 61 amino acids from the end of the protein.	[45]	7581404
c.1229T > G	p.Leu410Arg	Missense	FCT	No data	CM983928	hmz	G29	The substitution exhibits a shift in polarity from non-polar to positively charged and displays a decrease in Kyte–Doolittle hydrophobicity from 3.8 to −4.5. The variant occurs 57 amino acids from the end of the protein.	[46]	9762612
c.1261-1G > A	p.?	Splice site	FCT	No data	CS1913889	hmz	G30	This substitution affects the invariant AG acceptor splice site of intron 7.	[47]	31064022
c.1279C > T	p.Gln427*	Nonsense	FCT	<0.0001	CM890049	hmz; htz comp	G31 G3.2	This nonsense substitution truncates the protein at codon 427, which is 40 amino acids from the end of the protein.	[28,48]	2642067 8399358
c.1399T > A	p.(*467Lysext*79)	Stop-loss	na	No data	CM085972	htz	G24.2	The substitution causes an extension of the protein after codon 466.	[42]	18504684
c.194G > A	p.Gly65Asp	Missense	GHS	No data	CM930254	hmz	G32	The substitution exhibits a shift in polarity from non-polar to negatively charged and displays a decrease in Kyte–Doolittle hydrophobicity from −0.4 to −3.5. The variant occurs 402 amino acids from the end of the protein.	[33]	8504303
Exon 4 del	na	Gross del	GHS	No data	CG994924	hmz	G33	Loss of protein or enzymatic function.	[16]	10094192
Exon 7-8 del	na	Gross del	na	No data	CG910663	hmz	G34	Loss of protein or enzymatic function.	[1]	2012122
Entire gene del	na	Gross del	na	No data	CG994923	htz comp	G7.2	Loss of protein or enzymatic function.	[32]	10496076

^1^ genotype corresponds in Table 2 phenotype description, ^2^ nomenclature according HGVS v2.0 recommendations: RefSeq: NM_000147.4, NP_000138.2. FCT—α-L-fucosidase, C-terminal; GHS, glycoside hydrolase superfamily domain; HGMD, human gene mutation database; hmz, homozygote; htz comp, compound heterozygote; MAF, minor allele frequency according ExAc, 1000G, gnomAD databases; na, not applicable; null, no protein product or lack it’s activity; PMID, PubMed unique identifier; pts, patients; del, deletion; fs, frame shift.

**Table 2 genes-11-01383-t002:** Clinical characteristics of patients with fucosidosis, reported with novel pathogenic variants in the *FUCA1* gene.

Genotype Set ^1^	*FUCA1* Pathogenic Variant(s) and Zygosity		Ethnic Origin (pts)	Gender/No of Cases	Family History	Phenotype			Ref.
						Developmental History	Additional Clinical and Enzymatic Findings	Disease Type	
G1	c.203C > T, p.Ser68Leu	hom	Italian (1)	NA					[26]
G2	c.244C > T, p.Gln82*	hmz	Italian (1)						[27]
			Turkish (1)	4-year-old: male	Consanguineous parents (first degree cousins)	Delayed development—started to sit unassisted after one year, began to walk at age of 27 months, could not speak any meaningful words. Recurrent upper respiratory tract infections at four months (treated like an asthma patient, with no success). Mild gingival hypertrophy, bilateral ptosis, angiokeratomas on all over the trunk and scrotum and hypertonicity on lower extremities.	X-ray: mild form of dysostosis multiplex. Brain MR: combination of hypointensity in the medial and lateral pallidal segments of the globus pallidus and hyperintensity in its laminae on T2-W. MRS: spectral peaks at 3.8–3.9 ppm as well as a doublet at 1.2 ppm that inverts on TE 135.	Not classified	[28]
G3	c.244C > T, p.Gln82*/c.1279C > T, p.Gln427*	htz comp	N/A (2)	N/A					[29]
G4	c.355_364del10, p.(Glu119Thrfs*11)	hmz	Austrian (1)						[30]
G5	c.393T > A, p.Tyr131*	hmz	Chinese (1)	Male	Consanguineous parents (first-degree	Mild hypertrophic cardiomyopathy, mild aortic stenosis.	Recurrent bronchopneumonia and partial lung collapse (with chronic inflammatory process in high-resolution computerized tomography of the thorax).	Intermediate	[31]
			Taiwanese (1)	Female	Non-consanguineous parents		α-fucosidase activity markedly decreased in peripheral blood leukocytes (2.4 nmol/h per mg protein; control 24–162), while that of cultured skin fibroblasts even lower (0.24 nmol/h per mg protein; control 96–360); α-fucosidase activity conspicuously decreased in peripheral blood leukocytes (0.5 nmol/h/mg protein; normal range 50–200 nmol/h/mg protein).	Late infantile-onset with slow progressive symptoms, considered to be type I	[17]
G6	c.437delC, p.(Pro146Argfs*41)	hmz	Italian (8)	NA					[27]
G7	c.459G > A, p.Trp153*/ entire gene del, loss of protein	htz comp	Japanese (1)	Female	Non-consan-guineous parents	At 23 months: delayed speech and hearing difficulty. At 3 years and 7 months: coarse face, small stature and kyphoscoliosis. At 4 years: angiokeratoma on the palms, bone abnormalities and motor dysfunction gradually progressed. At age 6 years: unable to walk, myoclonic seizures developed. At age 13 years: spasticity and dystonia of all extremities with involuntary, movements, generalized angiokeratoma corporis diffusum, no hepatosplenomegaly or pubertal development.	α-L-fucosidase in leukocytes (0 nmol/h per mg protein compared with 29.1 ± 4.7 nmol/h per mg protein in control).	Chronic, slow progressive	[32]
G8	c.464C > T, p.Ser155Phe/c.790C > T, p.Arg264*	htz comp	Spanish or Portuguese (1)	NA					[33]
G9	c.467_468delAA, p.(Lys156Argfs*11)/c.1160G > A, p.Trp387*	htz comp	Italian (1)	NA			All patients had negligible enzyme activity and reduced CRIM.	Not classified	[34]
G10	c.525-76_663-163del3282, p.?/c.671delC, p.(Pro224Leufs*3)	htz comp	Thai (1)	Male	Non-consanguineous parents	Until 2 years of age: milestones reportedly attained within normal limits. At age 3 years and 6 months: psychomotor regression (lost his ability to communicate verbally, had spasticity in which the lower extremities were more affected than the upper). Growth parameters and development continued to deteriorate At 7 years: bed-ridden, coarse facial features with macroglossia. At age 9 years: angiokeratomas.	Brain MRI: increasing degrees of cerebral atrophy with significant signal changes in the thalamus. Chest X-ray: oar-like ribs, bullet-shaped vertebrae, and widening of both clavicular heads.	Not classified	[35]
G11	c.564G > A, p.Trp188*	hmz	Austrian (1)	NA					[30]
G12	c.648C > A, p.Tyr216*	hmz	Belgian (1)						[27]
G13	c.661delA, p.(Ser221Alafs*6)	hmz	British (1)						[27]
G14	c.661delA, p.(Ser221Alafs*6)/unknown variant ^2^	htz comp	Canadian-Indian (1)						[30]
G15	c.717C > A, p.Tyr239*	hmz	Chinese (1)	4-year-old male (Patient 2)	Consanguineous marriage family	Short stature, seizure, psychomotor delay, mild scoliosis facial dysmorphism (frontal bossing, epicanthus, low nasalbridge, long philtrum, and thick lips), increased alkaline phosphatase; died at the age of 6 years.	Almost no expression of *FUCA1* mRNA.	Atypical fucosidosis type I	[36]
G16	c.768+1G > A, p.?	hmz	Syrian (1)	NA			Leukodystrophy	Not classified	[37]
G17	c.773delA, p.(Glu258Glyfs*3)	hmz	Turkish (1)	Male	Consanguineous (second or third cousin) parents; younger brother died of the same disease at the age of 10	Diagnosed at the age of 1 year, because of progressive neurological regression, spasticity, contractures, coarse face, hepatosplenomegaly, growth retardation, and angiokeratoma corporis diffusum; no speech. Bedridden until his death at the age of 22.	Very low CRIM for α-L-fucosidase (1.5% of normal mean) and negligible enzyme activity measured with 4-methylumbelliferyl α-L-fucoside as a substrate in cultured skin fibroblasts	Not classified	[38]
G18	c.810delC, p.(Cys271Valfs*59)	hmz	Portuguese (1)			NA			[27]
G19	c.837_838delTG, p.Cys279*	hmz	Iranian (1)	4-year-old girl (and her uncle, aged 23 years)	Consanguineous parents (second cousins)	Proband: – neonatal period: low blood glucose and seizures until 1.5 years of age: developmental milestones reportedly within normal limits then: gradual loss of verbal communication, eating and walking deterioration 4 years: abnormal bone development, spasticity and no eye contact, hepatomegaly. Proband’s uncle: disease had started at the age of 5 years old and deteriorated gradually until he was completely paralyzed and bedridden at presentation.		I in proband II in her uncle	[39]
G20	c.969+1G > A, p.?	hmz	East Indian-Zambian (1)	Female	A first cousin of the proband died aged 4 years, was reported to have had identical physical and developmental problems	From 18 months: gradually became uninterested in toys, started mouthing and casting objects, and language skills deteriorated to grunts and pointing; gait became progressively more unsteady—walked on a broad gait and fell backwards after every few steps; aged 9 years: hypoacusis (hearing thresholds raised at 50 dB bilaterally with flat tympanograms) wore a helmet to prevent head injury and arm splints to deter hand chewing.	Weight and height below the 3rd centile, head circumference on the 5th centile; coarse facial features, protruding tongue, kyphosis, contracture of the right elbow, and hirsutism no activity of α-L-fucosidase in the leucocytes.	Not classified	[40]
G21	c.1000A > T, p.Asn334Tyr	hmz	Austrian (1)	N/A					[30]
G22	c.1003dupT, p.(Tyr335Leufs*9)	hmz	Sudanese (1)		No data, only “In some cases, it was possible to attribute this homozygosity to consanguinity or to substantiate it by family studies.”	N/A	All (6) patients had been diagnosed as having fucosidosis by the finding of decreased α-L-fucosidase activity in white blood cells, fibroblasts, or serum to confirm a preliminary clinical diagnosis.	Severe (4) Not known (1) Moderate/ severe (1)	[30]
G23	c.1030_1095dup, p.(Asp344_Asn365dup)	hmz	N/A (1)		Not presented	N/A	Lymphoid cells lacked α-L-fucosidase activity but contained a reduced amount (<5% of control) of an immunoreactive α-L-fucosidase.	Not classified	[41]
G24	c.1034G > A, p.Gly345Glu/ c.1399T > A, p.(*467Lysext*79)	htz comp	N/A (1)	Female	Non-consanguineous parents	Very slow psychomotor development with episodes of developmental stagnation or mild regression and recovery, muscular hypotonia as the predominant clinical sign. From 24 months: progressive neurological deterioration At age of 48 months: unable to sit unaided, truncal hypotonia, brisk deep tendon reflexes and increasing spasticity of the limbs with evolving contractures, atypical absences.	α-L-fucosidase in plasma: 0.00 mU/mL (reference range: 2.00–9.99), in leukocytes 0.01 mU/mg (reference range: 46–179). Brain MR—severe hypomyelination and the sign of T2 hyperintense curvilinear streaks separating the lentiform nucleus into three partitions, corresponding to the lateral and medial medullary laminae of the globi pallidi and reflecting non-myelinated fibers.	Not classified	[42]
G25	c.1060delA, p.(Arg354Glyfs*9)	hmz	no data (1)	NA					[43]
G26	c.1138G > T, p.Glu380*	hmz	Hispanic-American (8)						[44]
G27	c.1160G > A, p.Trp387*	hmz	no data (1)						[27]
G28	c.1216G > T, p.Gly406*	hmz	German (2)	Males	Consanguineous parents (third cousin) parents in both patients	Patient AB—diagnosed with fucosidosis at the age of 2.1 years. Patient SW—diagnosed with fucosidosis at the age of 2.3 years. Both exhibited slowly progressive neurological deterioration by the age of 3 years.	α-L-fucosidase activities in the leukocytes negligible.	Not classified	[45]
G29	c.1229T > G, p.Leu410Arg	hmz	No data (1)	Female	Consanguineous parents	(Described initially at the age of 20 years by Primrose et al. in 1975 [50] with “progressive physical and mental retardation, short stature, angiokeratoma corporis diffusum, dysostosis multiplex and generalized muscle wasting.”) At the age of 46: lost all verbal and most nonverbal communication, nonambulant and unable to sit unaided; had suffered continued muscle wasting, several minor long-bone fractures and one chest infection. weight—20 kg height—113 cm; generalized muscle wasting and flexion contractures of the legs with prominent scoliosis; intermittent protrusion and writhing of the tongue and a mucopurulent discharge in the mouth and external nares; prominent angiokeratomas on the thighs, legs, and trunk and a network of fine capillaries on the limbs.		Not classified	[46]
G30	c.1261-1G > A, p.?	hmz	No data (1)	8-year-old boy	Consanguineous parents	From the age of 2 years— gradual loss of motor and speech skills, and recurrent chest infections. By 4 years of age— legs spasticity with ankle clonus and equinovarus deformity of the feet, the spasticity involved the upper extremities over the following year, leaving him with spastic tetraparesis. By the age of 6 years— dystonic posturing of the legs, and gradually spread to involve the trunk and the facial muscles over the next 6 months. At the age of 8 years— short stature and facial dysmorphism (wide mouth, thick lips, and misaligned incisors with attrition), angiokeratoma lesions over the neck, upper chest, lower back, and finger tips.	Low activity of α-fucosidase in leukocytes (0.18 nmol/h/mg protein; ref. range: 19–266.6 nmol/h/mg protein) and a positive qualitative urine based thin layer chromatography test.	Not classified	[47]
G31	c.1279C > T, p.Gln427*	hmz	Italian (2) Cuban (2) French (2)	N/A	N/A				[48]
G32	c.194G > A, p.Gly65Asp	hom	French-American (3pts) Italian (4pts)	N/A					[33]
G33	exon 4 deletion	hom	Dutch (2)	Male					[16]
G34	exon 7-8 del	hmz	Algerian (2)	N/A					[1]

^1^ The detailed description of genotype data are in Table 1, ^2^ The variant in the other allele is not known; N/A, not available, CRIM, cross-reactive immunological material.

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
