# Peer review of "Fucosidosis—Clinical Manifestation, Long-Term Outcomes, and Genetic Profile—Review and Case Series"

_genes, 2020, doi:10.3390/genes11111383_

Round 1
Reviewer 1 Report
The authors Stepien Ciara and Jezela-Stanek present a complete and interesting review of fucosidosis.
Fucosidosis is an extremely rare disease and I have to admit that I haven't seen any patient with fucosidosis.
The manuscript is quite complete.
My only comments refer to the introduction that I found not very clear: Lines 31 -33 need to be clarified. Authors do not explicit clearly the role of fucosidase. The lines 31-33 are confusing and may be replaced later.
The manuscript is very interesting and give a complete overview of the disease.
Author Response
The authors Stepien Ciara and Jezela-Stanek present a complete and interesting review of fucosidosis.
Fucosidosis is an extremely rare disease and I have to admit that I haven't seen any patient with fucosidosis.
The manuscript is quite complete.
My only comments refer to the introduction that I found not very clear: Lines 31 -33 need to be clarified. Authors do not explicit clearly the role of fucosidase. The lines 31-33 are confusing and may be replaced later.
RESPONSE: It has now been clarified and a sentence was added:
‘As a result of the hydrolytic enzyme deficiency, incomplete catabolism of N- and O-glycosylproteins results in the accumulation …’
The manuscript is very interesting and give a complete overview of the disease.
RESPONSE: Thank you
Reviewer 2 Report
This is a detailed review of an ultra-rare disease, fucosidosis. Some considerations:
- You say in abstract: Fucosidosis is a lysosomal storage disease (LSD) is a neurodegenerative disorder which 13 progresses inexorably. But this is not well expressed, it should be reviewed
- t is also said: Only 36 pathogenic variants in the FUCA1 gene are associated with fucosidosis. this is not correct, it must be said at the current time or date
- It would also be positive to comment on the abstract about the genotype-phenotype relationship
- Some reference number like number 4 is not well expressed
Author Response
This is a detailed review of an ultra-rare disease, fucosidosis. Some considerations:
- You say in abstract: Fucosidosis is a lysosomal storage disease (LSD) is a neurodegenerative disorder which 13 progresses inexorably. But this is not well expressed, it should be reviewed
RESPONSE: It has now been corrected to:
‘Fucosidosis is a neurodegenerative disorder which progresses inexorably’.
- It is also said: Only 36 pathogenic variants in the FUCA1 gene are associated with fucosidosis. this is not correct, it must be said at the current time or date
RESPONSE: It is correct and it is further explained in line 75-76:
‘Only 36 pathogenic variants in the FUCA1 gene reported in HGMD are associated with fucosidosis, so far (www.hgmd.cf.ac.uk ; updated November 10, 2020) [25] (Table 1).’
- It would also be positive to comment on the abstract about the genotype-phenotype relationship
RESPONSE: The sentence has been added in the abstract:
‘The genotype-phenotype correlation has not been well defined.’
- Some reference number like number 4 is not well expressed
RESPONSE: It has now been corrected in line 29.
Round 2
Reviewer 2 Report
The suggested modifications have been made and I accept the article